# Pharmacist-Initiated Pre-Emptive Pharmacogenetic Panel Testing with Clinical Decision Support in Primary Care: Record of PGx Results and Real-World Impact

**DOI:** 10.3390/genes10060416

**Published:** 2019-05-29

**Authors:** Cathelijne H. van der Wouden, Paul C. D. Bank, Kübra Özokcu, Jesse J. Swen, Henk-Jan Guchelaar

**Affiliations:** 1Department of Clinical Pharmacy and Toxicology, Leiden University Medical Center, 2333 ZA Leiden, The Netherlands; 2Leiden Network for Personalised Therapeutics, 2333 ZA Leiden, The Netherlands; 3Division of Pharmacoepidemiology and Clinical Pharmacology, Utrecht Institute for Pharmaceutical Sciences (UIPS), Utrecht University, 3584 CG Utrecht, The Netherlands

**Keywords:** pre-emptive, pharmacogenetics, panel

## Abstract

Logistics and (cost-)effectiveness of pharmacogenetic (PGx)-testing may be optimized when delivered through a pre-emptive panel-based approach, within a clinical decision support system (CDSS). Here, clinical recommendations are automatically deployed by the CDSS when a drug-gene interaction (DGI) is encountered. However, this requires record of PGx-panel results in the electronic medical record (EMR). Several studies indicate promising clinical utility of panel-based PGx-testing in polypharmacy and psychiatry, but is undetermined in primary care. Therefore, we aim to quantify both the feasibility and the real-world impact of this approach in primary care. Within a prospective pilot study, community pharmacists were provided the opportunity to request a panel of eight pharmacogenes to guide drug dispensing within a CDSS for 200 primary care patients. In this side-study, this cohort was cross-sectionally followed-up after a mean of 2.5-years. PGx-panel results were successfully recorded in 96% and 68% of pharmacist and general practitioner (GP) EMRs, respectively. This enabled 97% of patients to (re)use PGx-panel results for at least one, and 33% for up to four newly initiated prescriptions with possible DGIs. A total of 24.2% of these prescriptions had actionable DGIs, requiring pharmacotherapy adjustment. Healthcare utilization seemed not to vary among those who did and did not encounter a DGI. Pre-emptive panel-based PGx-testing is feasible and real-world impact is substantial in primary care.

## 1. Introduction

An individual’s response to a drug can be predicted by their pharmacogenetic (PGx) profile [1,2]. Incorporation of an individual’s PGx profile into drug prescribing promises a safer, more effective and thereby more cost-effective drug treatment [3,4]. Several randomized controlled trials (RCTs) demonstrate the clinical utility of pre-emptive single gene tests to guide dosing [5,6,7], and drug selection [8], for individual drug-gene interactions. These studies are perceived as a proof-of-concept supporting the clinical utility of pre-emptive PGx testing, and may therefore also be applied to other drug-gene interactions, for which evidence of the same rigour may lack [9,10]. The Dutch Pharmacogenetics Working Group (DPWG) was established in 2005 to devise clinical guidelines for individual drug-gene interactions based on a systematic review of literature [11,12]. These guidelines provide clinicians with recommendations on how to manage drug-gene interactions. To date, the DPWG has developed guidelines for 97 drug-gene interactions, of which 54 are actionable drug-gene interactions, many of which are encountered principally in primary care. In parallel, the Clinical Pharmacogenetics Implementation Consortium (CPIC) has also devised guidelines for more than 40 drugs [13]. The DPWG and CPIC guidelines have ongoing efforts to harmonize the two [14]. In the Netherlands, the DPWG guidelines are incorporated into a nationwide clinical decision support system, called the “G-standaard”, providing pharmacists and general practitioners (GPs) with relevant clinical recommendations at the point of care when an actionable drug-gene interaction is encountered.

Significant debate persists regarding the optimal approach for implementing PGx testing in clinical care; where some support using a pre-therapeutic single gene approach and others a pre-emptive panel-based approach [15]. The pre-therapeutic single gene approach has several drawbacks. In this one-at-a-time strategy, an individual gene is tested in response to a first prescription of an interacting target drug. If, however, patients receive prescriptions for multiple interacting target drugs over time, they may require testing for multiple single genes. Here, pharmacotherapy may be delayed in awaiting the PGx results. Furthermore, the costs of single gene testing may be allocated a multitude of times, while the marginal cost of testing and interpreting additional pharmacogenes simultaneously is near-zero [16,17]. These logistical and cost-effectiveness issues may be overcome and optimized when delivering PGx in a panel-based approach [18]. Here, a panel of variants within multiple genes, which are associated with drug response, are tested and saved for later use in preparation of future prescriptions [15]. In this way, the panel-results can be reused over time, as multiple drugs which interact with multiple variants are prescribed [19]. When an interacting target drug is prescribed, the corresponding PGx guideline can be deployed by the clinical decision support system at the point of care, thereby providing clinicians with the necessary information to guide prescribing by PGx, without any delay. Alternatively, a combination of the two strategies may be the optimal approach for delivering PGx. Here, a panel test is ordered reactively in response to an incident prescription and is saved in the electronic medical record (EMR) for pre-emptive use in future prescriptions. However, in order for the clinical decision support system to be enabled, it is crucial that the PGx results are recorded and preserved in the EMR. If this fails, a potential drug-gene interaction may go unnoticed. As a result, the added value of testing multiple genes is lost. A recent study showed that PGx results for *CYP2D6* were sparsely recorded; only 3.1% and 5.9% of reported PGx results were recorded in EMRs by general practitioners (GPs) and pharmacists, respectively, within a mean follow-up of 862 days [20]. This indicates that correct record of PGx results in the EMR may be a remaining barrier preventing the realization of panel-based testing. However, this is yet undetermined when reporting the results for multiple genes simultaneously. Therefore, we sought to investigate whether pharmacists and GPs are able to record PGx panel testing results within their EMR, in order to enable life-long use of PGx results through a clinical decision support system.

Another barrier preventing implementation of panel-based PGx testing is the lack of evidence demonstrating its clinical utility. Although there is a firm evidence base supporting the clinical utility of pre-emptive single gene PGx testing, evidence of similar quality supporting a panel-based approach is lacking [21]. Even so, several smaller studies report promising results indicating that pre-emptive panel-based PGx guided prescribing is indeed (cost-)effective in preventing adverse drug reactions among polypharmacy and psychiatry patients. However, this is yet to be determined within primary care [22,23,24,25,26,27]. Alternatively, the clinical impact of population-wide panel-based testing has previously been modelled by using Medicare prescription data; indicating half of patients above 65 will use at least one of the drugs for which PGx guidelines are available during a four year period, and one fourth to one third, will use two or more of these drugs [28]. Another study showed that more than 60% of the population would benefit from PGx guided prescribing within a 5-year period [19]. However, the clinical impact is yet undetermined in a real-world setting. This may differ from modelled estimations since the patients selected by pharmacists to receive panel testing may differ from those included in prescription datasets. Therefore, we aim to quantify the potential real-world impact of implementation of PGx panel in a clinical decision support system within a side-study of the Implementation of Pharmacogenetics into Primary care Project (IP3 study). In this side-study, the primary outcome is the frequency at which patients receive newly initiated prescriptions, with possible drug-gene interactions, for which PGx results are available in the EMR. To explore which target groups may benefit most from panel testing, we aim to investigate which patient sub-groups may more frequently initiate newly prescribed drugs within follow-up. Secondary outcomes include their downstream impact on healthcare utilization. Firstly, we hypothesize that patients who encounter an actionable drug-gene interaction and adhered to the DPWG guidelines will have a similar healthcare utilization compared to those who did not encounter an actionable drug-gene interaction. Secondly, we hypothesize that patients who encounter an actionable drug-gene interaction, but did not adhere to the DPWG guidelines, have a higher healthcare utilization compared to those who did not encounter an actionable drug-gene interaction.

## 2. Materials and Methods

### 2.1. Study Design, Participants

We performed a cross-sectional follow-up of The Implementation of Pharmacogenetics into Primary care Project (IP3 study) cohort, as a side-study. The IP3 study is a prospective multicenter observational pilot study with the objective to test the feasibility of pharmacist-initiated pharmacogenetics testing within a clinical decision support system in primary care. The study design, rationale and main study findings have previously been described elsewhere [29]. In brief, community pharmacies in the vicinity of Leiden, The Netherlands, were invited to participate in the study. Pharmacists who agreed on participation were provided with the opportunity to request free PGx tests for a panel of 40 variants in eight pharmacogenes (see Appendix A), to guide drug dispensing based on the DPWG guidelines, for a maximum of 200 patients. The genes selected to be tested were based on genes for which DPWG guidelines are available and which are either included in the Affymetrix Drug Metabolizing and Transporters (DMET) array (*CYP2C9*, *CYP2C19*, *CYP2D6*, *CYP3A5*, *SLCO1B1*, *TPMT* and *VKORC1*) or determined in clinical care (*DPYD*). This panel can be used in combination with the DPWG guidelines to guide drug prescribing for 41 drugs. Here, a combination of reactive and pre-emptive panel testing is implemented. A PGx panel is ordered reactively in response to an incident prescription and is saved in the EMR for pre-emptive use is future prescriptions. Adult patients receiving a first prescription (defined as no prescription for the first drug within the preceding 12 months) for at least 28 days for one of 10 drugs (amitriptyline, atomoxetine, atorvastatin, (es)citalopram, clomipramine, doxepin, nortriptyline, simvastatin or venlafaxine) in routine care were eligible. Additional in- and exclusion criteria are reported elsewhere [29]. After identification of the patients through automated queries, the participating pharmacists manually checked whether patients fulfilled the in- and exclusion criteria. Finally, patients not recruited within 14 days after dispensing the first prescription were excluded. When patients were eligible, pharmacists were able to select these patients for ordering a PGx panel. The panel test result could be used reactively for the drug of enrolment and pre-emptively for future prescriptions of 41 drugs with potential drug-gene interactions.

### 2.2. Healthcare Setting

In the Dutch healthcare system, patients are typically listed with one GP and one pharmacy. The GP plays a gatekeeping role in the provision of healthcare. The GP is consulted for all initial healthcare problems and may refer to specialized care when appropriate. Typically, GPs maintain EMRs for their patients and contain prescription history, lab results, correspondence with specialized physicians and reports regarding ER (emergency room) visits and hospitalizations. In parallel, pharmacists maintain a separate EMR containing dispensing history, relevant contra-indications and drug allergies and are used for medication surveillance at drug dispensing.

In routine care, PGx testing is predominantly performed within hospital pharmacy or clinical chemistry laboratories. Hospitals additionally maintain a separate EMR for registered patients. Generated PGx results are typically recorded in the hospital’s EMR and are communicated with requesting pharmacists of physicians in primary care by paper or electronic reports.

### 2.3. Ethics Approval

All subjects gave their written informed consent for enrolment before they participated in the study. The study was conducted in accordance with the Declaration of Helsinki, and the protocol was approved by the Ethics Committee of Leiden University Medical Center (LUMC) (P14.081). Patients provided informed consent for data collection regarding their medication and related outcomes from both pharmacy and GP EMRs within 3 years of enrolment.

### 2.4. DNA Collection, Isolation, Extraction and Genotyping

After providing signed informed consent, pharmacists collected a 2mL saliva sample from participating patients using the Oragene DNA OG-250 (DNA Genotek Inc). The samples were transported to the PGx laboratory in Leiden University Medical Center by research staff or mail. DNA was extracted in accordance to Oragene DNA OG-250 isolation procedure, where a solution volume of 100µL, instead of 200 µL, was used. The DNA concentration was quantified in each sample with NanoDropPhotometer (Thermo Fisher Scientific), and DNA quality was assessed with the use of the 260 nm/280 nm absorbance ratio. Genotypes of *CYP2C9*, *CYP2C19*, *CYP2D6*, *CYP3A5*, *DPYD*, *SLCO1B1*, *TPMT* and *VKORC1* were determined using the Drug Metabolizing and Transporters (DMET) Plus Array (Affymetrix, Santa Clara, CA). *CYP2D6* copy number variants were detected with qPCR (Thermo Fisher Scientific, Massachusetts, USA). The DMET array was supplemented with the *DPYD* 1236G>A and 2846A>T variants which were routinely tested in clinic at the LUMC. Validation of the assays is described elsewhere [29].

### 2.5. Translation of Genotype to Phenotype and Return of Results

Genotypes for the eight pharmacogenes were translated into predicted phenotypes using the DPWG guidelines. A paper report holding the genotypes, predicted phenotypes and the DPWG therapeutic recommendation for the drug of enrollment was devised and sent to the patients’ general practitioner (GP) and pharmacist by mail and/or fax (see Appendix A for an example report). The report held the request to record the entire PGx profile in the EMR to enable the clinical decision support system when drug-gene interaction is encountered during drug prescribing or dispensing (see Figure 1). Predicted phenotypes must be recorded in the EMR in a contra-indication format to enable deployment of the relevant guideline through the clinical decision support system. Even if patients are predicted to be extensive metabolizers (EM), we recommend that they still be recorded as contra-indications to record the performance of this test. However, pharmacy EMRs can hold a maximum of 10 contra-indications. It is important to note that the pilot study is initiated through the pharmacists and therefore the GPs who receive the paper report may have had no prior knowledge about the existence of the IP3 pilot study.

### 2.6. Healthcare Provider Incorporation of PGx Results in Drug Prescribing and Dispensing

When an actionable drug-gene interaction is encountered, the DPWG guideline directs adjustment of drug, dose or vigilance of pharmacotherapy to avoid potential adverse drug reactions or lack of efficacy. However, pharmacists are free to choose whether to adhere to the DPWG guidelines. In The Netherlands, and within the IP3 study, pharmacists must discuss pharmacotherapy alteration, resulting from medication surveillance, with the prescribing physicians before the prescription can be altered.

### 2.7. Groups for Analysis

Patients have been stratified into three groups for comparison (see Table 1): 1) those who did not encounter an actionable drug-gene interaction for the drug of enrolment, 2) those who encountered an actionable drug-gene interaction for the drug of enrolment and whose health care providers chose to adhere to the DPWG guideline, and 3) those who encountered an actionable drug-gene interaction for the drug of enrolment and whose health care providers chose not to adhere to the DPWG guideline.

### 2.8. Outcomes and Analyses

In this side-study, the primary outcome for quantifying the feasibility of the panel-based approach is whether the PGx panel results were recorded as a contra-indication in both the GP and pharmacist EMRs at the time of follow-up.

In this side-study, the primary outcome for quantifying the real-world impact of the panel-based approach is the number of newly initiated drugs for which potential drug-gene interactions are encountered, since enrolment, and whether these interactions are actionable. A potential drug-gene interaction is encountered when a patient, regardless of their phenotype (e.g., *CYP2D6* PM, IM or EM), receives a new prescription for a drug for which an actionable DPWG guideline is available and the interacting gene was included in the IP3 panel (e.g., metoprolol-*CYP2D6* guideline). A potential drug-gene interaction becomes an actionable when the patient’s predicted phenotype directs adjustment of pharmacotherapy, based on the relevant DPWG guideline (e.g., patient is *CYP2D6* PM and initiates metoprolol). See Appendix A for a list of drugs for which actionable DPWG guidelines are available and IP3 panel results can be used to identify potential and actionable drug-gene interactions. To explore which target group may benefit most from panel testing, we investigate whether baseline demographic variables (gender, age, BMI, number of comorbidities and number of comedications) are associated with an increasing number of prescribed drugs with potential drug-gene interactions within follow-up by using univariate negative binomial regression. The secondary outcome is healthcare utilization as a result of pre-specified drug-gene interaction associated adverse drug reactions within 12 weeks of enrolment. This is a composite endpoint of GP consults (in person, by phone or by e-mail), emergency department (ED) visits, and hospitalizations. These drug-gene interactions associated adverse drug reactions were defined before data collection was initiated and are based on the literature underlying the DPWG guidelines. For example, if a patient enrolled on simvastatin with a *SLCO1B1* TC genotype consults their GP regarding muscle pain symptoms within 12 weeks of initiation, this is considered a drug-gene interaction associated adverse drug reactions since *SLCO1B1* TC and CC carriers are at higher risk for statin-induced myopathy [30]. See Appendix A for an overview of pre-specified drug-gene interaction associated adverse drug reactions and underlying literature. We compare the frequency of the composite endpoint among patients who encounter an actionable drug-gene interaction and adhered to the DPWG guidelines (group 2) to those who did not encounter an actionable drug-gene interactions associated adverse drug reactions(group 1), using binomial logistic regression in a non-inferiority analysis. We have set a non-inferiority at a margin of 1.2. Secondly, we compare the frequency of the composite endpoint among patients who encounter an actionable drug-gene interaction, but did not adhere to the DPWG guidelines (group 3), to those who did not encounter an actionable drug-gene interaction (group 1), using binomial logistic regression.

## 3. Results

### 3.1. IP3 Cohort and Follow-Up

Overall 200 patients were enrolled in the IP3 study between November 2014 and July 2016. Patient characteristics are presented in Table 1. The database containing the genotypes and predicted phenotypes is available at https://databases.lovd.nl/shared/individuals (patient IDs 184080-184279). 62 (31.0%) patients encountered an actionable drug-gene interaction for the drug of enrolment, as previously reported by Bank et al. [29]. Of these, health care providers chose to adhere to the DPWG guideline in 49 (79.0%) cases. Data collection was performed retrospectively between April 2018 and September 2018 in both pharmacy and GP EMRs; from pharmacy EMRs between May 4th 2018 and May 29th 2018; and from GP EMRs between April 3rd 2018 and September 28th 2018. Data could be retrospectively collected cross-sectionally from 200 (100%) and 177 (88.5%) pharmacy and GP EMRs, respectively (see Figure 2). The mean follow-up from pharmacy EMRs was 933 days (range 649–1279), approximately 2.5 years. The mean follow-up from GP EMRs was 917 days (range 622–1238).

### 3.2. Feasibility: Record of PGx Panel Results in the Pharmacy and GP EMRs

Record of PGx panel results by both pharmacists and GPs are shown in Figure 3. Pharmacists were able to record predicted phenotypes (including EMs) in 96.0% (*n* = 192) of pharmacy EMRs. In all cases they were recorded as contra-indications (100%, *n* = 192). Pharmacists failed to document the PGx results in 4.0% of cases (*n* = 8). The most common reason for failure of documentation (2.0%, *n* = 4) was merely due to PGx paper reports being lost in the pharmacy. The second most common reason was that the individual did not carry any aberrant variant, and was therefore predicted wildtype for all genes; this was the case for three patients (1.5%, *n* = 3). Pharmacists, therefore, felt it was not necessary to record EM phenotypes. Only one set of PGx results was failed to be documented in the EMR since the pharmacist did not know how to (0.5%). A discrepancy between the reported results and documented results was found in the records of two patients (1.0%). This was due to a manual error on account of the pharmacist.

General practitioners were able to record the PGx results in 67.8% (*n* = 120) of patient records. Of these, 34% (*n* = 59) were recorded as contra-indications and 35% (*n* = 61) in another format such as a PDF file.

### 3.3. Real-World Impact: Frequency of Newly Prescribed Drugs for Which PGx Results Were Available in the EMR

Table 2 shows the frequency of newly initiated drugs for which there were potential drug-gene interactions and PGx results were available in the EMR. 97.0% (*n* = 194) of patients received at least one subsequent drug for which PGx results were in the EMR. Within the follow-up time, a mean of 2.71 drugs for which the PGx results were available were prescribed, of these 0.66 (24.2%) were actionable drug-gene interactions, requiring pharmacotherapy adjustment. The most commonly prescribed drugs for which PGx results were available were atorvastatin (14.4%), simvastatin (9.4%) and pantoprazole (9.4%). The most common drugs which were actionable drug-gene interactions, however, were atorvastatin (28.2%), metoprolol (13.0%) and amitriptyline (8.4%). To explore who may benefit most from PGx-panel testing, Table 3 presents baseline demographics stratified by an increasing number of newly initiated drugs for which there were potential drug-gene interaction. It seems that the number of newly initiated prescriptions increases with age, number of comorbidities and number of comedications, but this could not be statistically concluded.

### 3.4. Real-World Impact: Downstream Effects of Actionable Drug-Gene Interactions on Healthcare Utilization

Table 4 shows that patients who encountered an actionable drug-gene interaction and whose health care providers adhered to the DPWG guidelines had a similar healthcare utilization as a result of a drug-gene interactions associated adverse drug reaction (40.0%) to those who did not carry an actionable drug-gene interaction (30.0%). This in line with our initial hypothesis. The 95%-CIs of the incidence of composite endpoint drug-gene interactions associated adverse drug reaction of groups 1 and 2 overlap. We therefore observe that there is no difference between the two groups. However, we cannot demonstrate non-inferiority since the upper limit of the 95%-CI of the OR of group 1 is not lower than the non-inferiority margin of 1.2.

We observed a much lower healthcare utilization as a result of a drug-gene interactions associated adverse drug reactions among patients carrying an actionable drug-gene interaction but whose health care providers did not adhere to the DPWG guidelines (0.0%) to those who did not carry an actionable drug-gene interaction (30.0%). This is in contrast to our initial hypothesis.

## 4. Discussion

We report what is, to our knowledge, the first assessment of the real-world impact of pharmacist-initiated pre-emptive panel-based testing in primary care. This side-study demonstrates that recording of PGx panel results in the EMR is feasible and enables health care providers to (re)use these results to inform pharmacotherapy of newly initiated prescriptions. 96% of PGx panel results were successfully recorded in the pharmacy EMR, enabling 97% of patients to (re)use these results for at least one, and 33% of patients for up to four newly initiated prescriptions, within a relatively short 2.5-year follow-up. Of all newly initiated prescriptions with a potential drug-gene interaction (*n* = 541), 24.2% (*n* = 131) were actionable drug-gene interactions, requiring pharmacotherapy adjustment. We expect the potential impact of pre-emptive panel-based testing to further increase with time as the likelihood of additional subsequent prescriptions increases.

With their dedication to medication surveillance, pharmacists are leading candidates to manage requesting of PGx testing, recording of PGx results and application of the PGx guidelines. This is confirmed by other pilot studies performed in pharmacy settings [31,32,33,34,35]. However, we found that both pharmacists and GPs are very able to record PGx results in their EMRs as contra-indications (96% and 33% of pharmacists and GPs, respectively); enabling deployment of relevant guidelines by the clinical decision support system when a drug-gene interaction is encountered both at prescribing and dispensing. An advantage of applying this double-verification is the minimization of the risk of missing a drug-gene interaction. As a result, it is not disastrous that GPs also recorded them in other formats, thereby not enabling the clinical decision support system at prescribing, in 35% of cases. In contrast, a recent study showed that genotyping results were sparsely communicated and recorded correctly; only 3.1% and 5.9% of reported genotyping results were recorded by GPs and pharmacists, respectively, within a similar follow-up time [20]. The discrepancy between these could be due to the pilot study setting or differences in PGx reporting methods. IP3 study researchers have visited the participating IP3 pharmacies multiple times within the follow-up period; possibly unintentionally reminding or motivating pharmacists to record PGx results, which they may otherwise have not performed. However, it is important to note that GPs were outside the scope of the pilot study setting, as they were not the enrolling health care providers, and therefore provide a less biased perspective on recording frequency. Still, it is much higher than that reported by Simoons et al. [20]. Surprisingly, 1.5% of PGx results were not recorded by pharmacists because they did not include actionable genotypes. However, it is still of importance to document these results to avoid unnecessary re-testing of the patient. Finally, the fact that discrepancies between reported results and the recorded result were only observed in 1% of pharmacy EMR cases, indicates that the current manual system of recording is error prone. Regardless of the low error rate, PGx results are static and therefore life-long. It is therefore imperative that errors in the recording of PGx results are avoided. Future initiatives should focus on the development of automated sharing of PGx results across EMRs. In the Netherlands, such an initiative has been the launched but requires patient consent before it can be utilized. The National Exchange Point (“Landelijk Schakel Punt” (LSP)) is a nationwide secured EMR infrastructure to which nearly health care providers access [36]. Only when a patient has provided written consent for the LSP, can a professional summary of the local pharmacy or GP EMR, including PGx results, be downloaded by another treating health care provider in the same region; unless the patient chose to shield this information. Alternatively, providing the PGx results directly to patients may resolve the issue in terms of communicating and recording PGx results; for example, utilizing the Medication Safety-Code card [37,38].

In the face of a time in which health care providers are confronted with an increasing number of variables to optimize clinical decision making, it is of utmost importance that this information is presented in a structured fashion; this is achieved by a clinical decision support system [39,40]. PGx testing results differ from other laboratory testing results because they remain applicable over a patient’s lifetime. We have demonstrated that, even within a relatively short follow-up, the real-world impact of a panel-based approach combined with a clinical decision support system is immense; almost all (97%) of patients used PGx results for at least one, and 33% of patients for up to four prescriptions within a relatively short 2.5-year follow-up. Of these, 24.2% (*n* = 131) were actionable drug-gene interactions. Similar proportions of actionable drug-gene interactions in primary care were found by Bank et al. (unpublished) [41]. Here, investigators overlaid the frequencies of phenotypes as observed within the IP3 cohort with nationwide prescription data spanning one year and found that 3.6 million incident prescriptions encountered a potential drug-gene interactions and of these, 856,002 (23.6%) encountered an actionable drug-gene interaction [41]. We observed drugs for which results were useful; these were primarily statins and proton pump inhibitors. This finding is in accordance with Samwald et al. [28]. The observed frequencies of potential drug-gene interactions, however, are much higher than reported by others previously [19,28]. Samwald et al. indicated half of the patients above 65 will use at least one of the drugs for which PGx guidelines are available during a four year period, and one fourth to one third will use two or more of these drugs [28]. Schildcrout et al. reported that 60% of the population would benefit from PGx guided prescribing within a 5-year period [19]. The higher frequency we observed could be a result of different target populations and drugs. Our sample consisted of patients selected by pharmacist and who initiated one of ten drugs, and therefore at higher risk for initiating subsequent drugs. Several promising studies indicate the effectiveness and effect of PGx panel-based testing on healthcare utilization in psychiatry and polypharmacy [22,23,24,25,26,27]. For example, Brixner et al. studied the effect of panel-based PGx testing with 6 genes on the healthcare utilization within polypharmacy patients. Results showed that the PGx screened cohort had a lower rate of ER visits (RR = 0.29, 95% confidence interval (CI) = 0.15–0.55, *p* = 0.0002) and a lower rate of hospitalizations (relative risk (RR) of 0.61, 95% CI = 0.39–0.95, *p* = 0.027). With this decrease in ER visits and hospitalizations, the authors concluded that PGx panel-based testing could potentially lead to cost-savings [23]. These cost savings may be potentially higher than that observed in primary care since polypharmacy patients have a higher a priori risk of hospitalization, as it increases with the number of comedications [42]. In this study we aimed to assess the downstream effects of an actionable drug-gene interaction on healthcare utilization. Although we did not observe a statistically significant difference between groups 1 (40%) and 2 (30%), we were not able to conclude non-inferiority, since this is a side-study by design and therefore was underpowered for a non-inferiority analysis. In contrast to our initial hypothesis we observed a much lower healthcare utilization among group 3 (0%) patients when compared to group 2 (30%). However, this cannot be concluded, since the adherence rate of HCPs was high, thereby resulting in a relatively low number of patients carrying an actionable DGI but whose HCPs did not adhere to the DPWG guidelines. Another limitation to this analysis is the retrospectively collected data from GP EMRs, which is prone to reporting bias. Nonetheless, gold-standard evidence demonstrating (cost-)effectiveness of this approach is required to convince stakeholders of population-wide implementation. An RCT aiming to generate such evidence is underway [21].

However, questions regarding who should be tested, and when it is most cost effective to perform pre-emptive panel testing, remain unanswered. In this side-study, we have chosen to perform pre-emptive panel testing among those who received a first prescription for one of ten drugs. Here, there is an initial delay of PGx testing results for the first prescription, but PGx results can be used uninterrupted, if recorded in the EMR, when future drug-gene interactions are encountered. On the one hand, it may be more cost-effective to perform population-wide testing at birth, to ensure maximization of instances in which a PGx result is available when a drug-gene interaction is encountered. In contrast to our approach, not one prescription will be delayed as a result of PGx testing. On the other hand, some may never encounter drug-gene interactions, thereby unintentionally wasting resources on PGx testing. To shed light on this issue, some have predicted which patients may benefit from PGx testing in the near future algorithmically and using prescription data [43,44]. Others have modelled the cost-effectiveness of testing a 40-year old for life-long prevention of adverse drug reactions using a Markov model [45]. Overall, a consensus has not been reached regarding whom and when to test [16]. Within this side-study we observe the number of newly initiated prescriptions, and thus potential benefit of panel testing, increases with age, number of comorbidities and number of comedications, although this was not statistically significant. However, since 97% of this cohort made re-use of their panel results, we may conclude that the in- and exclusion criteria of this study may be successful criteria in selecting patients who will further benefit from panel testing. The most cost-effective target groups applicable for panel testing must be further investigated.

In addition to unanswered timing and application of testing, the variants selected to be included in a PGx panel also require additional curation. Recently, the DPWG has provided a suggested panel (van der Wouden et al., unpublished) [46]. Here, variants included in the panel reflect the entire set of existing DPWG guidelines and are continuously updated as the field of PGx expands. It will be of utmost importance to record the version number of the tested panel, so that it can be retrieved which variants were tested within a specific gene. Moreover, the most cost-effective technique used to determine the PGx profile is also undetermined. As the cost of next-generation sequencing decreases, we envision a future in which we may be able to extract relevant PGx variant alleles from sequencing data [47], possibly making genotype based testing redundant. If this is to come into fruition, the determining the cost-effectiveness of implementing PGx testing will become redundant, as the information on PGx variants become secondary findings, free of additional costs. In this case, only effectiveness will be of interest. Overall, the cost-effectiveness of a panel-approach is a dependant on many variables including the target population, timing, tested variants and testing technique.

## 5. Conclusions

Both pharmacists and GPs are very able to record PGx results into their respective EMRs, thereby maximizing the potential benefits of PGx results when deployed by the clinical decision support system in future prescriptions. Within this cohort, almost all patients were able to benefit from the availability of the PGx-panel results in their EMR, indicating that the real-world impact of a panel approach is immense. The downstream impact on healthcare utilization was unable to be concluded due to the small sample size. Ongoing research will quantify the effects of pre-emptive panel-based testing on patient outcomes [21]. Future research should focus on assessing the most cost-effective approach regarding timing, target population, variants and techniques for PGx testing. Regardless, we argue that in terms of logistics, delivery through a clinical decision support system is most feasible.

## Figures and Tables

**Figure 1 genes-10-00416-f001:**
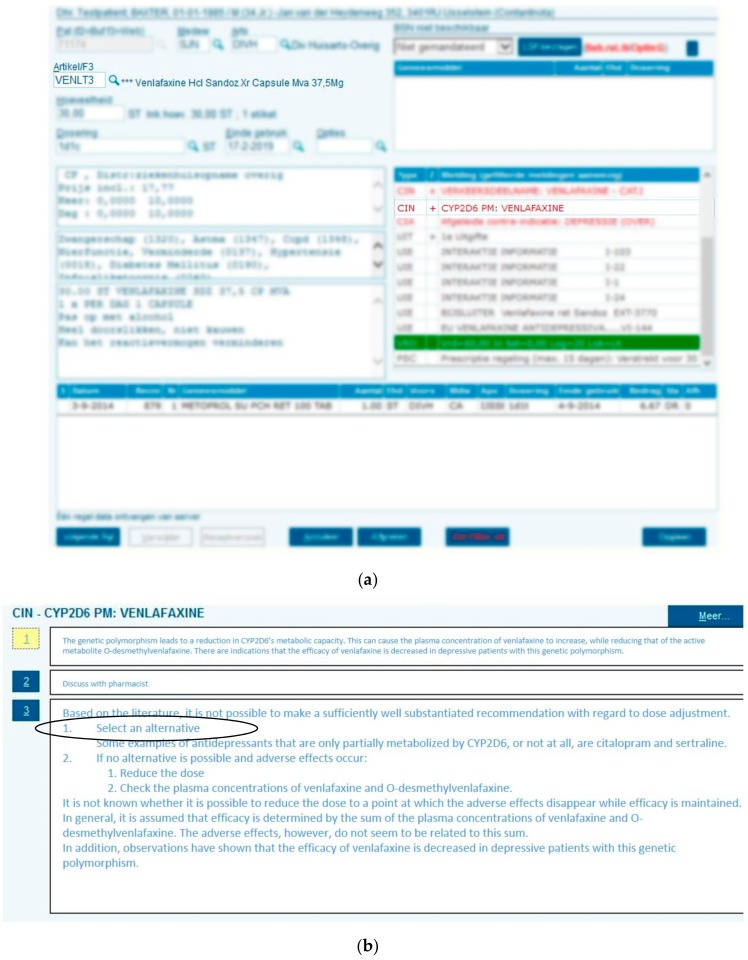
Clinical decision support during drug dispensing. A patient who is *CYP2D6* PM (as noted in the electronic medical record (EMR) as contra-indication, as indicated by “CIN” (contra-indication) receives a prescription for venlafaxine (**a**) which triggers a pop-up with the relevant Dutch Pharmacogenetics Working Group (DPWG) recommendation directing selection of alternative drug (**b**).

**Figure 2 genes-10-00416-f002:**
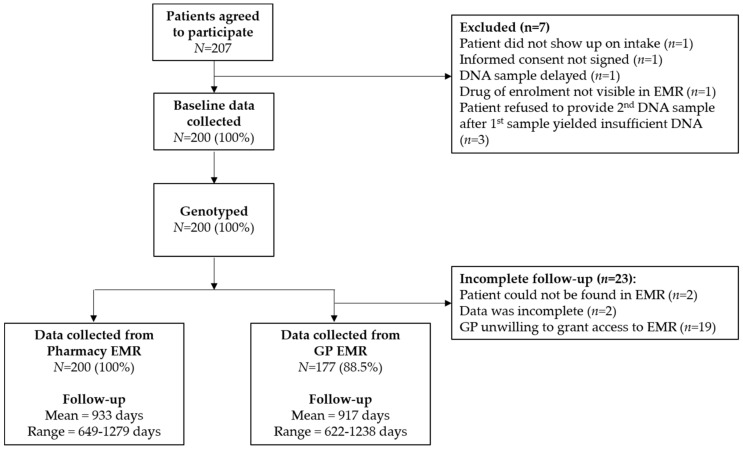
Flow chart or IP3 participant enrolment and follow-up.

**Figure 3 genes-10-00416-f003:**
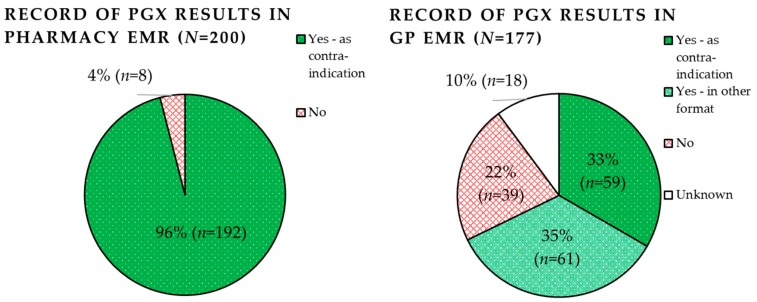
Record of pharmacogenetic panel results in the pharmacy and general practitioner (GP) electronic medical records (EMRs).

**Table 1 genes-10-00416-t001:** Summary of patient characteristics in Implementation of Pharmacogenetics into Primary care Project (IP3) cohort stratified by groups for analysis

	Overall IP3 Study Cohort (*n* = 200)	Groups for Analysis		
			Actionable Drug-Gene Interaction for the Drug of Enrolment (*n* = 62, 31.0%)	
		1) No Drug-Gene Interaction for the Drug of Enrolment (*n* = 138, 69.0%)	2) Health Care Provider Adhered to DPWG Guideline (*n* = 49, 24.5%) *	3) Health Care Providers did not Adhere to DPWG Guideline (*n* = 9, 4.5%) *
**Gender**				
Female, *n* (%)	103 (51.5)	74 (53.6)	25 (51.0)	3 (33.3)
Male, *n* (%)	97 (48.5)	64 (46.4)	24 (49.0)	6 (66.8)
**Age in years, Mean (SD)**	61.6 (11.2)	62.3 (11.0)	60.9 (11.5)	56.8 (13.3)
**BMI (kg/m^2^), Mean (SD)**	28.3 (14.9)	28.9 (17.7)	27.1 (4.5)	27.4 (2.4)
**Self-reported ethnicity father, *n* (%)**				
Caucasian	187 (93.5)	128 (92.8)	47 (95.9)	9 (100.0)
Other	13 (6.5)	10 (7.2)	2 (4.1)	0 (0.0)
**Self-reported ethnicity mother, *n* (%)**				
Caucasian	188 (94.0)	129 (93.5)	47 (95.9)	9 (100.0)
Other	12 (6.0)	9 (6.5)	2 (4.1)	0 (0.0)
**Drug of enrolment, *n* (%)**				
Amitriptyline	15 (7.5)	9 (6.5)	5 (10.2)	0 (0.0)
Atorvastatin	115 (57.5)	80 (58.0)	28 (57.1)	5 (55.6)
Citalopram	7 (3.5)	5 (3.6)	1 (2.0)	0 (0.0)
Escitalopram	3 (1.5)	2 (1.4)	1 (2.0)	0 (0.0)
Nortriptyline	17 (8.5)	10 (7.2)	5 (10.2)	2 (22.2)
Simvastatin	29 (14.5)	26 (18.8)	2 (4.1)	1 (11.1)
Venlafaxine	14 (7.0)	6 (4.3)	7 (14.3)	1 (11.1)
**Number of comorbidities at baseline, Mean (SD) ****	4.6 (2.5)	4.4 (2.4)	4.9 (2.6)	4.4 (2.3)
**Number of comedications at baseline, Mean (SD) ****	4.0 (3.3)	3.93 (3.4)	4.0 (2.9)	4.4 (3.0)

IP3: Implementation of Pharmacogenetics into Primary care Project; SD: standard deviation; BMI: body mass index; * Excluding others (*n* = 4): Recommendation given after drug was discontinued (*n* = 1); same dose (*n* = 1); dose increased and ECG unknown (*n* = 1); no drug-gene interaction and no action (*n* = 1). ** Based on *n* = 177 for whom data collection from GP records was completed.

**Table 2 genes-10-00416-t002:** Frequency of newly initiated drugs for which there were potential drug-gene interactions in subsequent prescriptions after pharmacogenetics panel in 200 primary care patients with a mean follow-up of 933 days (=2.56 years).

	Number of Patients (%)	Three Most Commonly Prescribed with Potential Drug-Gene Interaction, N (%)	Actionable Drug-Gene Interaction (%)	Three Most Commonly Prescribed with Actionable Drug-Gene Interactions, N (%)
**Subsequent drug 1**	194 (97%)	1. atorvastatin, 69 (35.6%) 2. omeprazole, 26 (13.4%) 3. pantoprazole, 20 (10.3%)	47 (24.2%)	1. atorvastatin, 19 (40.4%) 2. amitriptyline, 11 (23.4%) 3. citalopram, 6 (12.8%)
**Subsequent drug 2**	166 (83%)	1. atorvastatin, 32 (19,3%) 2. metoprolol, 29 (17.5%) 3. simvastatin, 21 (12.7%)	46 (27.7%)	1. atorvastatin, 14 (30.4%) 2. metoprolol, 10 (21.7%) 3. codeine, 6 (13.0%)
**Subsequent drug 3**	115 (57.5%)	1. pantoprazole, 20 (17.4%) 2. omeprazole, 19 (16.5%) 3. simvastatin, 15 (13.0%)	23 (20.0%)	1. metoprolol, 7 (30.4%) 2. simvastatin, 4 (17.4%) 3. codeine/venlafaxine, 3 (13.0%)
**Subsequent drug 4**	66 (33%)	1. simvastatin, 15 (22.7%) 2. pantoprazole, 11 (16.7%) 3. atorvastatin, 9 (13.6%)	15 (22.7%)	1. atorvastatin, 4 (26.7%) 2. venlafaxine/simvastatin/ clopidogrel, 2 (13.3%) 3. citalopram/omeprazole/codeine/ flecainide/metoprolol, 1 (6.7%)
**Overall**	**541**	**1. atorvastatin, 78 (14.4%)** **2. simvastatin, 51 (9.4%)** **3. pantoprazole, 51 (9.4%)**	**131 (24.2%)**	**1. atorvastatin, 37 (28.2%)** **2. metoprolol, 17 (13.0%)** **3. amitriptyline, 11 (8.4%)**
**Mean per patient (SD)**	**2.71 (1.1)**	**-**	**0.66 (0.8)**	**-**

SD: standard deviation.

**Table 3 genes-10-00416-t003:** IP3 cohort stratified by number of newly initiated drugs with a potential drug-gene interaction within follow-up.

	Overall IP3 Study Cohort (*n* = 200)	0 (*n* = 6, 3%)	1 (*n* = 27, 13.5%)	2 (*n* = 52, 26%)	3 (*n* = 50, 25%)	≥4 (*n* = 65, 32.5%)	*p*-Value *
**Gender**							0.775
Female, n (%)	103 (51.5)	4 (66.7)	12 (44.4)	24 (46.2)	27 (54.0)	36 (55.4)	
Male, n (%)	97 (48.5)	2 (33.3)	15 (55.6)	28 (53.8)	23 (46.0)	29 (44.6)	
**Age in years, Mean (SD)**	61.6 (11.2)	53.3 (16.3)	59.4 (10.6)	61.0 (11.5)	63.0 (10.5)	62.8 (11.1)	0.442
**BMI (kg/m** **^2^** **), Mean (SD)**	28.3 (14.9)	25.6 (2.6)	29.1 (5.8)	27.4 (4.5)	27.6 (4.8)	29.6 (25.2)	0.854
**Number of comorbidities at baseline, Mean (SD) ****	4.6 (2.5)	3.4 (1.1)	4.0 (2.2)	4.0 (2.5)	4.6 (2.3)	5.4 (2.6)	0.232
**Number of comedications at baseline, Mean (SD) ****	4.0 (3.3)	3.0 (2.1)	3.4 (3.4)	3.3 (3.4)	3.8 (2.7)	5.1 (3.4)	0.279

SD: standard deviation; BMI: body mass index; * Univariate negative binomial regression; ** Based on *n* = 177 for whom data collection from GP records was completed.

**Table 4 genes-10-00416-t004:** Healthcare utilization as a result of drug-gene interaction associated adverse drug reactions within 12 weeks of enrolment.

	Overall IP3 Study Cohort *n* = 200		Actionable Drug-Gene Interaction for the Drug of Enrolment	
		1) No Drug-Gene Interaction for the Drug of Enrolment *n* = 138 (69.0%)	2) Health Care Provider Adhered to DPWG Guideline *n* = 49 (24.5%)	3) Health Care Provider did not Adhere to DPWG Guideline *n* = 9 (4.5%)
GP EMR follow-up completed (%)	177 (88.5%)	120 (87.0%)	45 (91.8%)	8 (88.9%)
Number of patients experiencing drug-gene interactions associated adverse drug reactions	56 (31.6%)	37 (30.8%)	19 (43.2%)	0 (0.0%)
**Composite endpoint drug-gene interactions associated adverse drug reactions**				
Number of patients, n (%)	**54 (30.5%)**	**36 (30.0%)**	**18 (40.0%)**	**0 (0.0%)**
95% CI		**66.0%–80.6%**	**47.1%–73.7%**	
**GP consults as a result of drug-gene interactions associated adverse drug reactions**				
Number of patients, n (%)	52 (29.4%)	35 (29.2%)	17 (37.8%)	0 (0.0%)
Number of GP consults, Mean (SD)	53, 2.19 (2.11)	35, 2.06 (1.99)	18, 2.44 (2.36)	0, 0 (0)
**ER visit as a result of drug-gene interactions associated adverse drug reactions**				
Number of patients, n (%)	3 (1.7%)	1 (0.8%)	2 (4.4%)	0 (0%)
Number of ER visits, Mean (SD)	3, 1 (1)	1, 1 (1)	2, 1 (1)	0, 0 (0)
**Hospitalization as a result of drug-gene interactions associated adverse drug reactions**				
Number of patients, n (%)	1 (0.6%)	1 (0.6%)	0 (0.0%)	0 (0.0%)
Number of hosp., Mean (SD)	1, 1 (1)	1, 1 (1)	0, 0 (0)	0, 0 (0)
**Binomial Logistic Regression (group 1 and 2)**				
**OR [95%CI] ***		**1.81 [0.89, 3.67]**		

GP: general practitioner; OR: odd ratio; CI: confidence interval; * Including gender, age, and BMI as covariates.

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
