# Peer review of "Pharmacist-Initiated Pre-Emptive Pharmacogenetic Panel Testing with Clinical Decision Support in Primary Care: Record of PGx Results and Real-World Impact"

_genes, 2019, doi:10.3390/genes10060416_

Round 1

Reviewer 1 Report

The aim of the manuscript of van der Wouden et al., is represented by the establishment of the feasibility and the real-world impact of a pre-emptive panel-based pharmacogenetic test in primary care. 

The study took advantage from the organisation of the  healthcare system in The Netherlands and involved pharmacists (also deputed to collect biological samples) and general practitioners. The study enrolled 200 patients and results showed that pre-emptive panel-based PGx-testing is feasible and the real-world impact is substantial in primary care. 

This is an excellent study. It is well designed and methodologies used are fully adequate. The analysis of a large panel of SNPs was performed by using a standardised and accurate array. From a practical point of view, the analysis of ADME SNPs allowed patients to further use pharmacogenetic results for subsequent prescriptions. Physicians and pharmacists were provided with a synthetic but punctual report. 

MINOR COMMENTS

The screenshot reported in Figure 1 is in Dutch and, however, Figure 1B in unreadable. If possible, it would be better to translate the Figure in English, or at least to obtain a Figure with a higher resolution.

Author Response

Reviewer 1:

The aim of the manuscript of van der Wouden et al., is represented by the establishment of the feasibility and the real-world impact of a pre-emptive panel-based pharmacogenetic test in primary care.

The study took advantage from the organisation of the healthcare system in The Netherlands and involved pharmacists (also deputed to collect biological samples) and general practitioners. The study enrolled 200 patients and results showed that pre-emptive panel-based PGx-testing is feasible and the real-world impact is substantial in primary care.

This is an excellent study. It is well designed and methodologies used are fully adequate. The analysis of a large panel of SNPs was performed by using a standardised and accurate array. From a practical point of view, the analysis of ADME SNPs allowed patients to further use pharmacogenetic results for subsequent prescriptions. Physicians and pharmacists were provided with a synthetic but punctual report.

MINOR COMMENTS

The screenshot reported in Figure 1 is in Dutch and, however, Figure 1B in unreadable. If possible, it would be better to translate the Figure in English, or at least to obtain a Figure with a higher resolution.

We thank the reviewer for the suggestion. To comply with the reviewer’s suggestion we have faded out unnecessary text in Figure 1A, and have translated of relevant sections of Figure 1B. We were also able to increase the size of Figure 1B to improve its readability.

Reviewer 2 Report

van der Wouden et. al resubmit their study entitled "Pharmacist-initiated pre-emptive pharmacogenetic panel testing with clinical decision support in primary care: record of PGx results and real-world impact."

The authors have now included the requisite information that allows a reader the opportunity to distinguish this analysis from the primary study. They also removed the genotyping success rate data, which is an improvement.

It is still problematic to cite unpublished work, so I will make the following practical recommendations:

112 The study design, rationale and main study findings have previously been described elsewhere (unpublished) [29]. - Cut this sentence altogether, and add any details you need.

126 Additional in- and exclusion criteria are reported elsewhere [29]. - Cut this sentence and put the criteria in the supplement

162 Validation of the assays is described elsewhere [29]. - Cut this sentence, it is unnecessary.

229 enrolment, as previously reported by Bank et al. [29]. Cut, "as previously reported by Bank et al. [29]." You may need to add "as previously reported" to the Bank et. al manuscript if this manuscript is published first.

Author Response

Reviewer 2:

van der Wouden et. al resubmit their study entitled "Pharmacist-initiated pre-emptive pharmacogenetic panel testing with clinical decision support in primary care: record of PGx results and real-world impact."

The authors have now included the requisite information that allows a reader the opportunity to distinguish this analysis from the primary study. They also removed the genotyping success rate data, which is an improvement.

It is still problematic to cite unpublished work, so I will make the following practical recommendations:

112 The study design, rationale and main study findings have previously been described elsewhere (unpublished) [29]. - Cut this sentence altogether, and add any details you need.

126 Additional in- and exclusion criteria are reported elsewhere [29]. - Cut this sentence and put the criteria in the supplement

162 Validation of the assays is described elsewhere [29]. - Cut this sentence, it is unnecessary.

229 enrolment, as previously reported by Bank et al. [29]. Cut, "as previously reported by Bank et al.

[29]." You may need to add "as previously reported" to the Bank et. al manuscript if this manuscript is published first.

We thank the reviewer for these valuable suggestions. We agree that citing unpublished work in this context is problematic and would require the adjustments as proposed by the reviewer. However, we expect the publication of Bank et al. in due course. Therefore, once Bank et al. is published, we will replace [29] with the citation referring to Bank et al. in the manuscript proofs, as discussed with the Genes editorial team.

Reviewer 3 Report

The authors present findings of a study aiming to quantify the potential impact of implementation of PGx panel in a  clinical decision support system. 

They hypothesized that patients who encounter an actionable drug-gene interaction (DGI) and adhered to the DPWG guidelines will have a similar healthcare utilization compared to those who did not encounter an actionable DGI. Secondly, they hypothesized that patients who encounter an actionable DGI but did not adhere to the DPWG guidelines have a higher healthcare utilization compared to those who did not encounter an actionable DGI. The authors found that healthcare utilization seemed not to vary among patients who did and did not encounter a DGI.

I have a few comments:

-      The selection of genes is unclear, more details should be provided, particularly because reference 29 is still unpublished.

-      The Introduction is at times verbose and should be made more concise. For instance in “Here, a panel test is ordered reactively in response to an indecent prescription and is saved in the EMR for pre-emptive use in future prescriptions.” it is not clear what the authors mean by using “indecent”

-      The sentence “Diplotypes for the eight pharmacogenes were translated into predicted phenotypes using the DPWG guidelines. “ is unclear, why just diplotypes?

-      The concept of non-inferiority should be used woth caution here as we are dealing with a retrospective analysis

-      This appears as a secondary analysis rather than a “side-study”

-      The use of abbreviations is too frequent and makes the manuscript difficult to follow

-      The discussion is way too optimistic “This side-study demonstrates that the 
 logistical pre-requisites to enable delivery of PGx results through a CDSS can be overcome in a pilot 
 setting and that the real-world impact of a panel approach is immense;” The authors should moderate their statement. In the end 
they did not find a specific impact of PGx testing. 

-      In the Results they write “However, this cannot be concluded as the adherence rate of HCPs was high, thereby resulting in a relatively low number of patients carrying an actionable DGI but whose HCPs did not adhere to the DPWG guidelines.” but this aspect should be treated in the Discussion

Author Response

Reviewer 3:

The authors present findings of a study aiming to quantify the potential impact of implementation of PGx panel in a  clinical decision support system.

They hypothesized that patients who encounter an actionable drug-gene interaction (DGI) and adhered to the DPWG guidelines will have a similar healthcare utilization compared to those who did not encounter an actionable DGI. Secondly, they hypothesized that patients who encounter an actionable DGI but did not adhere to the DPWG guidelines have a higher healthcare utilization compared to those who did not encounter an actionable DGI. The authors found that healthcare utilization seemed not to vary among patients who did and did not encounter a DGI.

I have a few comments:

-      The selection of genes is unclear, more details should be provided, particularly because reference 29 is still unpublished.

We thank the reviewer for this valuable suggestion. As described in line 117, the genes selected to be tested were based on those included in the Affymetrix DMET array (CYP2C9, CYP2C19, CYP2D6, CYP3A5, SLCO1B1, TPMT and VKORC1) and complimented with those determined in clinical care (DPYD). As a result of the reviewer’s comments we have added the following text to further explain gene selection:

Line 123 (in document with track-changes):

The genes selected to be tested were based on genes for which DPWG guidelines are available and which are either included in the Affymetrix DMET array (CYP2C9, CYP2C19, CYP2D6, CYP3A5, SLCO1B1, TPMT and VKORC1) or determined in clinical care (DPYD).

-      The Introduction is at times verbose and should be made more concise. For instance in “Here, a panel test is ordered reactively in response to an indecent prescription and is saved in the EMR for pre-emptive use in future prescriptions.” it is not clear what the authors mean by using “indecent”

We thank the reviewer for this valuable suggestion. As a result, we have reworded this section of the introduction (see below) and have corrected the typo “indecent” to “incident”.

Removal of “and PGx results are not applied to optimize drug therapy” in line 74 (in document with track-changes).

Removal of “in addition to the gene important to the incident prescription” in line 75 (in document with track-changes).

Removal of “such as would be the case after panel testing” in line 81 (in document with track-changes).

-      The sentence “Diplotypes for the eight pharmacogenes were translated into predicted phenotypes using the DPWG guidelines. “ is unclear, why just diplotypes?

We thank the reviewer for their suggestion and apologize for the confusion. As a result, we have reworded “diplotypes” to “genotypes” (line 171, in document with track-changes).

-      The concept of non-inferiority should be used with caution here as we are dealing with a retrospective analysis

We have chosen to perform a non-inferiority analysis since we are unable to conclude superiority based on the groups for analysis. Indeed, non-inferiority analyses require larger sample sizes than superiority analyses, for which we are underpowered in this case. Also, the retrospective collection of data induces reporting bias. We thank the reviewer for mentioning this issue and have as a result emphasized these two points in the discussion.

Lines 407-416 (in document with track-changes): In this study, we aimed to assess the downstream effects of an actionable drug-gene interaction on healthcare utilization. Although we did not observe a statistically significant difference between groups 1 (40%) and 2 (30%), we were not able to conclude non-inferiority, since this is a side-study by design and therefore was under powered for a non-inferiority analysis. In contrast to our initial hypothesis we observed a much lower healthcare utilization among group 3 (0%) patients when compared to group 2 (30%). However, this cannot be concluded, since the adherence rate of HCPs was high, thereby resulting in a relatively low number of patients carrying an actionable DGI but whose HCPs did not adhere to the DPWG guidelines. Another limitation to this analysis is the retrospectively collected data from GP EMRs, which is prone to reporting bias.

-      This appears as a secondary analysis rather than a “side-study”

We thank the reviewer for their valuable suggestion. However, the presented analyses were performed with data that was collected after the conclusion of the primary IP3 study. Although the same cohort was used, a new protocol was written for this study. We therefore feel it is better suited as a “side-study” rather than a secondary analysis, which implies it was included in the initial IP3 study protocol.

-      The use of abbreviations is too frequent and makes the manuscript difficult to follow

We thank the reviewer for their valuable suggestion and apologise for the inconvenience. We have removed the following abbreviations from the manuscript: DGI (drug-gene interaction), DGI-ADI (drug-gene interaction associated adverse drug reaction), HCP (health care provider) and CDSS (clinical decision support system).

We have chosen to keep the following abbreviations since they are either names or commonly used abbreviations: IP3 (Implementation of Pharmacogenetics into Primary care Project), EMR (electronic medical record), DPWG (Dutch Pharmacogenetics Working Group), ER (emergency room), GP (general practitioners), and PGx (pharmacogenetics).

-      The discussion is way too optimistic “This side-study demonstrates that the

 logistical pre-requisites to enable delivery of PGx results through a CDSS can be overcome in a pilot  setting and that the real-world impact of a panel approach is immense;” The authors should moderate their statement. In the end they did not find a specific impact of PGx testing.

We thank the reviewer for their comment and upon reflection agree that this closing statement is too optimistic. We have adjusted the statement to more reflect the main findings of the study (see below).

Line  327(in document with track-changes):

This side-study demonstrates that recording of PGx panel results is feasible and enables HCPs to (re)use these results to inform pharmacotherapy of newly initiated prescriptions. 96% of PGx panel results were successfully recorded in the pharmacy EMR, enabling 97% of patients to (re)use these results for at least one, and 33% of patients for up to four newly initiated prescriptions, within a relatively short 2.5-year follow-up.

-      In the Results they write “However, this cannot be concluded as the adherence rate of HCPs was high, thereby resulting in a relatively low number of patients carrying an actionable DGI but whose HCPs did not adhere to the DPWG guidelines.” but this aspect should be treated in the Discussion

We thank the reviewer for their comment and upon reflection agree that this statement is misplaced in the Results section. We have moved this statement to the Discussion section.

Line 411 (in document with track-changes): “In contrast to our initial hypothesis we observed a much lower healthcare utilization among group 3 (0%) patients when compared to group 2 (30%). However, this cannot be concluded as the adherence rate of HCPs was high, thereby resulting in a relatively low number of patients carrying an actionable DGI but whose HCPs did not adhere to the DPWG guidelines.”

Round 2

Reviewer 3 Report

No further comments

This manuscript is a resubmission of an earlier submission. The following is a list of the peer review reports and author responses from that submission.

Round 1

Reviewer 1 Report

van der Wouden and colleagues, deal with an interesting paper on Pharmacogenetics through a drug-gene interaction study translated in a real-world setting. Authors aimed at quantifying both feasibility and real-world impact of this approach in the primary care setting.

Although, potentially very attractive and interesting, a huge limit is the low number of enrolled population and the quite arbitrary sorting of drugs and genes included or not included.

Major points

1. An example to approach the study, and to improve the originality of the manuscript, could be that to compare molecules sharing the same pathology and drug-gene (e.g. the warfarin molecule). Authors included in their analysis “acenocoumarol”, and VKORC1 and CYP2C9 genes are considered in the presented manuscript. This is a good opportunity that should not be missed. They could discuss on warfarin versus acenocoumarol drug-gene interactions by their suggested pharmacogenetics approach (real-life translation).

These drugs are very commonly used worldwide with very few exceptions, and although they act by a similar mechanism, there are significant individual differences between them, especially in terms of their variability in response. This could be a pharmacogenetics example of drug-shifting to include and comment in the manuscript (additional practical examples would be extremely interesting also considering the translational medicine purpose).

Accordingly, Authors begin Introduction section stating “An individual’s pharmacogenetic (PGx) profile can be used to optimize drug and dose selection”, and additional concepts are afterward stressed as “Incorporation of PGx ….promises safer, …more cost-effective drug treatment” and more “..gene tests to guide dosing, and drug selection, for individual drug-gene interactions (DGIs)”.

2. Authors state in the manuscript “Unfortunately, the small sample size of this pilot study did not lend itself for statistical analysis of effects of encountering a (non-)adhered DGI on healthcare utilization”.

The lack of a complete and supportive statistical analyses is a strong limit that should be listed in a dedicated paragraph together with additional limitations.

3. Authors conclude the manuscript stating that “Future research should focus on assessing the most cost-effective approach regarding timing, target population, variants and techniques for PGx testing.”

Basically everybody can benefit of that approach, rather it could be of primary importance to find out and list the mutual scores of investigated populations (e.g. comparing women / men, and younger / elder people, etc.) also in terms of subset populations, and gender medicine in order to firstly mark those populations /patients really might take advantage.

Minor points

1. Ref. 29. Bank, P.; Swen, J.; Schaap, R.; Klootwijk, D.; Baak-Pablo, R.F.; Guchelaar, H.J. Implementation of pharmacist initiated pre-emptive pharmacogenomics testing in primary care (in submission). 2019.

Please remove from the list, rather comment in the test.

2. Supplementary Figure 1: Example report sent to physicians and pharmacists

It is worthless when written in local language

3. How Authors checked for potential fault in genotyping? Did they have internal controls? Were genotypes assessed in duplicate? Have they alternative back-up methodology?

4. What about warfarin PGx? Being a group of genes widely investigated and warfarin has a wide distribution worldwide (see above).

Author Response

Please see word document attached. 

Reviewer 2 Report

This study aims to quantify both feasibility and real-world impact of a pre-emptive panel-based approach to pharmacogenomics within a CDS system. There are issues with novelty, outcomes, and concomitant submissions. Still, the paper is well written with statistical rigor and may provide a foundation for future investigations. I'd like to give the authors the benefit of the doubt.

Major critiques:

The biggest critique is that this manuscript is thin on real data and meaningful outcomes. Testing the success rate of reports being uploaded to an EMR or the genotyping success rate push the tolerance of reader interest. The primary outcome of "the frequency at which patients receive newly initiated prescriptions, with possible DGIs, for which PGx results are available in the EMR" is unlikely to move the needle to show real world impact. This outcome could be modeled by simply looking at population-based prescription frequencies and allele frequencies. As such, this is an entirely descriptive paper of a prospective cohort. Nonetheless, if the authors can better demonstrate how this pilot study will build for future investigations, there may be some value.

There is second submission under review: Bank, P.; Swen, J.; Schaap, R.; Klootwijk, D.; Baak-Pablo, R.F.; Guchelaar, H.J. Implementation of pharmacist initiated pre-emptive pharmacogenomics testing in primary care (in submission). 2019. I am not sure why the adoption of the PGx testing by pharmacists needs to be decoupled from the primary outcome of this manuscript: successfully recorded in the pharmacy EMR. These two outcomes appear inter-related. Since the other manuscript "in submission" is referenced multiple times in the present manuscript, it is requested that the authors provide a working copy to review so the reviewers and editors can make informed assessments of the independence of the outcome presented. 

Please provide your power calculation for the primary outcome. Please provide p-values.

How were the 200 patients selected? How many were screened? What is the consent rate?

There is a disconnect between the introduction/discussion and methods. In the introduction and discussion, the authors create a binary comparison between a pre-therapeutic single gene approach to the pre-emptive panel based approach. In the methods, the authors state that subjects are eligible if they receive a first prescription of an actionable drug - this approach is not strictly preemptive and is more of a combination of the two approaches above. This disconnect (i.e. false rhetorical controversy) should be reconciled and the introduction should be revised. In reality, other implementation efforts involve pre-therapeutic testing in response to a single drug, but upload an entire panel for the future (See INGENIOUS trial, Eadon et al. CPT 2016). Thus, you may not be the first to use this approach as you contend. I would suggest refocusing the introduction and conclusion to better highlight the unique features of your implementation effort. Example: you may be the first to have pharmacists hand pick patients, why is this approach better?

Author Response

Please see word document attached.

Round 2

Reviewer 1 Report

I consider the Author's reply sufficiently responding to my requests.

Reviewer 2 Report

Upon reviewing both manuscripts:

1. Van der Wouden CH, Bank PCD, et. al. Piloting pharmacist-initiated pre-emptive pharmacogenetic panel testing with clinical decision support in primary care: feasibility and real-world impact 2019

and

2. Bank, P.; Swen, J.; Schaap, R.; Klootwijk, D.; Baak-Pablo, R.F.; Guchelaar, H.J. Implementation of pharmacist initiated pre-emptive pharmacogenomics testing in primary care (in submission). 2019.

It is evident that the overall project the authors have undertaken is laudatory. Implementation of pharmacogenomics is important and the authors have done so in a novel pharmacist led design. With that said, there exist issues with data-splitting, the definition of primary and secondary endpoints, and the duplication of endpoints. These issues preclude a complete understanding of the work undertaken and are likely to reduce a reader's comprehension of either manuscript alone. 

Major issues:

1. The definition of primary endpoint(s) is obscured. When I first read the van der Wouden et. al manuscript, I thought I was reading the primary outcomes of the IP3 cohort study and that reference 29 (Bank et. al) was a baseline study description. It is now clear that this is not the case. The van der Wouden manuscript reads as though it presents the primary outcome of the IP3 study and it does not mention any other primary endpoints. In the Bank et. al manuscript, "The primary endpoint was the adoption rate of PGx testing. This was defined as the number of genotyped patients divided by the total number of eligible patients in the timeframe of the study." In the Van der Wouden manuscript, the primary endpoint "is the frequency at which patients receive newly initiated prescriptions, with possible DGIs, for which PGx results are available in the EMR" and "whether PGx panel results were recorded as a contra-indication in both the GP and pharmacist EMRs at the time of follow-up." Each manuscript fails to mention the other (co-?)primary endpoints. In the present circumstance, this reviewer thinks that van der Wouden et. al are actually presenting secondary outcomes of the IP3 study (since the "main" study outcomes are referenced in Bank et. al); however, you have provided a power calculation, so it is also possible that these are co-primary endpoints. When a clinical trial (such as IP3) is conducted, primary outcomes should be pre--specified and co-primary endpoints should be relayed clearly so readers understand them. Generally, all co-primary endpoints are discussed in the same manuscript. If the van der Wouden manuscript were to be published now in its present form, leaving out other primary outcomes would not be ethical. Should the Bank article be published and the Wouden article address the definition of endpoints, this would solve the problem.

2. The authors state: "The paper by Bank et al. describes the adoption of PGx testing in primary care and only includes data collected at baseline. This manuscript, on the other hand, describes follow-up data collected from both pharmacist and GP EMRs and is thereby able to provide insight into clinical outcomes of the intervention." However, both manuscripts contain outcomes of up to 2.5 years with prescribing behavior - albeit different aspects of prescribing behavior. This is not wrong, but it is a form of data-splitting. This data-splitting makes the interpretation of each manuscript more complex, particularly because it is not clear what the actual primary goal of the study was, and which are simply secondary observations.

3. The DNA quality outcome is a secondary endpoint which is reported in the results of both manuscripts. This is duplication of an endpoint. This reviewer will acknowledge that there are subtle differences in the form of presentation because additional details are presented  in the Van der Wouden et. al article; however, at the core, this data is the same.

4. The Bank et. al paper (citation 29) is still unpublished. This reference is cited 5 times in the present manuscript and is vitally important for understanding what data is unique to the van der Wouden et. al paper. These citations and references should be removed. Without that paper published, the authors would need to include the following in this manuscript:

a. The design, rationale and main study findings.

b. The inclusion and exclusion criteria

c. The validation of the genotyping assay.

d. Clearly delineate the true primary and secondary endpoints of the IP3 study.

After reading both manuscripts, it seems like the van der Wouden study is a secondary study and should be re-submitted after publication of the Bank et. al study.